# A highly selective and recyclable NO-responsive nanochannel based on a spiroring opening—closing reaction strategy

Yao Sun[1], Sen Chen[1], Xiaoya Chen[2], Yuling Xu[1], Siyun Zhang[1], Qingying Ouyang[1], Guangfu Yang[1] & Haibing Li[1]

Endogenous nitric oxide (NO) is an important messenger molecule, which can directly activate $K^+$ transmission and cause relaxation of vascular smooth muscle. Here, inspired by the $K^+$ channel of smooth muscle cells, we report, a novel NO-regulated artificial nanochannel based on a spiro ring opening—closing reaction strategy. This nanofluidic diode system shows an outstanding NO selective response owing to the specific reaction between o-phenylenediamine (OPD) and NO on the channel surface with high ion rectification ratio (~6.7) and ion gating ratio (~4). Moreover, this NO gating system exhibits excellent reversibility and stability as well as high selectivity response. This system not only helps us understand the process of NO directly regulating biological ion channels, but also has potential application value in the field of biosensors.

---

[1] Key Laboratory of Pesticide and Chemical Biology (CCNU), Ministry of Education, International Joint Research Centre for Intelligent Biosensor Technology and Health, Chemical Biology Center, College of Chemistry, Central China Normal University, 430079 Wuhan, P.R. China. [2] State Key Laboratory of Chemo/Biosensing and Chemometrics, Hunan University, 410082 Changsha, P.R. China. Correspondence and requests for materials should be addressed to G.Y. (email: gfyang@mail.ccnu.edu.cn) or to H.L. (email: lhbing@mail.ccnu.edu.cn)

Nitric oxide (NO) is a highly reactive free radical that acts as a secondary messenger in signal pathways and an important regulatory molecule in many physiological processes, including regulation of blood pressure, immune response, cell adhesion, and neural communication[1–3]. It has been well documented that endogenous NO exerts many of its biological functions through interference with intracellular signaling pathways and regulates different ion fluxes in channels[4,5]. For example, NO can directly activate the gating of K+ channels in vascular smooth muscle[6]. Moreover, many diseases, including inflammation, have been linked to NO function abnormality[7,8]. Considering the significant role of NO in living systems, constructing an NO-regulated ion channel in vitro will help to better understand the mechanism of NO physiological functions. However, biological lipid bilayer ion channels are very instable and fragile in an external environment[9–12]. Therefore, developing an artificial ion channel to mimic the biological process of NO-regulated ion transport with excellent stability and robustness is crucial and highly demanded, and it will have a direct effect on the fields of biotechnology and materials science.

Functional single conical artificial nanochannels have been actively explored to mimic their bio-counterparts because of their excellent robust mechanical and chemical properties[13–22]. Moreover, functionalized nanochannels can effectively adjust ion current rectification and the gating states in response to external stimuli, such as pH, temperature, and specific ions, interacting with the ligand immobilized on the channel surface[23–33]. Up to now, most of the previously reported responsive nanochannels to regulate an ion (such as K+) are based on the non-covalent binding, such as electrostatic interactions, van der Waals forces, and hydrogen bonds between the external stimuli and ligand on the nanochannel surface[34–41]. For example, Hou et al. reported a functional nanochannel to regulate K+ based on the conformational change of the G-quadruplex DNA[34]. Ali et al. reported a pH-responsive nanochannel to regulate K+ based on the polyelectrolyte self-assembly inside the channel surface[35]. However, these non-covalent interactions may result in subtle interference with ligands on the channel surface and decrease the selectivity of channel regulation, especially for gas-driven systems[36]. Recently, our group reported a thiol-yne reaction-based nanochannel-responsive system that exhibits high cysteine specificity and selectivity even in complex matrices[42]. Despite this success, the thiol-yne strategy is essentially irreversible, resulting in recycling of this responsive system being impossible[43,44]. Therefore, developing a universal strategy for constructing a gas-driven nanochannel with high selectivity and recyclability is urgently required.

Herein, inspired by the NO-activated K+ channel in nature, we report the utilization of a reversible covalent reaction strategy to fabricate a nanochannel-responsive system that exhibits high NO specificity and stability (Fig. 1). In this system, the modified ligand is composed of two parts: o-phenylenediamine (OPD), which functions as a specific NO-reactive group, and rhodamine (R-110) as a spiro ring cyclization/opening switch to realize the reversibility of the system (Fig. 1)[45,46]. When spiro ring formation (cyclization) between OPD and R-110 occurs, the charge density of the channel surface become positive, resulting in inhibition of the ion transportation activity (OFF state). The ion transportation ability is triggered (ON state) by the NO-induced spiro ring opening and exposure of the −COOH groups of R-110. Therefore, this nanofluidic diode exhibits both ion gating and ion current rectification tuned by NO. Moreover, it involves specific cleavage of covalent bonds on the channel surface, thereby overcoming the low selectivity of the previous nanochannel-responsive systems and showing a highly specific and sensitive NO response without interference. The NO-driven device also

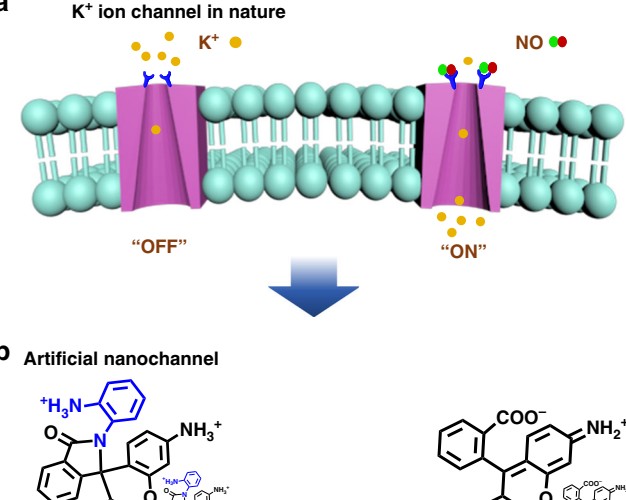

**Fig. 1** Schematic depiction of the ion channels. **a** NO-responsive ion channel in nature. **b** Scheme of the design of an NO-responsive nanochannel using a spiro ring opening/closing reaction strategy

exhibits excellent reversibility and stability, which indicates its efficiency and reusability for real-world applications, such as NO detection and biosensors.

## Results

**Fabrication of the NO-responsive nanochannel.** To produce the NO-responsive nanodevice, a single conical nanochannel was fabricated in a polyethylene terephthalate (PET, 12-μm thick) membrane using the well-developed track-etched technique. The large opening (base) was about 700 nm, as observed by scanning electron microscopy (Supplementary Fig. 1) and the tip was calculated to be 23 nm by an electrochemical method[47]. The interior surface of the nanochannel was modified by a two-step coupling reaction. First, rhodamine (R-110) was attached to the inner surface by the classical coupling reaction. OPD was subsequently conjugated with the −COOH groups of R-110 (Fig. 2a). Successful modification in each step was verified by the current−voltage ($I$−$V$) measurements in 0.1 M KCl solution. As shown in Fig. 2b, the $I$−$V$ characteristics of the nanochannel are significantly different before and after each modification step. The unmodified surface of the channel at a neutral pH (0.1 M KCl) is negatively charged owing to deprotonation of the −COOH groups, leading to ion current rectification. The decrease of the negative charge on the channel surface after immobilization of R-110 at the −COOH sites reduces the ion current rectification. After subsequent OPD conjugation with the −COOH groups of R-110, more −NH$_2$ groups were introduced by OPD, and protonation of the amino groups results in a positive charge on the surface of the channel, and immediate reversal of the rectifying characteristics (Fig. 2b). In addition, to further verify successful modification, the wettability and chemical composition of the nanochannels before and after modification were characterized by contact angle (CA) and X-ray photoelectron spectroscopy (XPS) measurements (Supplementary Figs. 2−3, Supplementary Tables 1−3). As expected, modification of the nanochannels with R-110 and OPD molecules leads to a remarkable change in the wettability of the surface (from

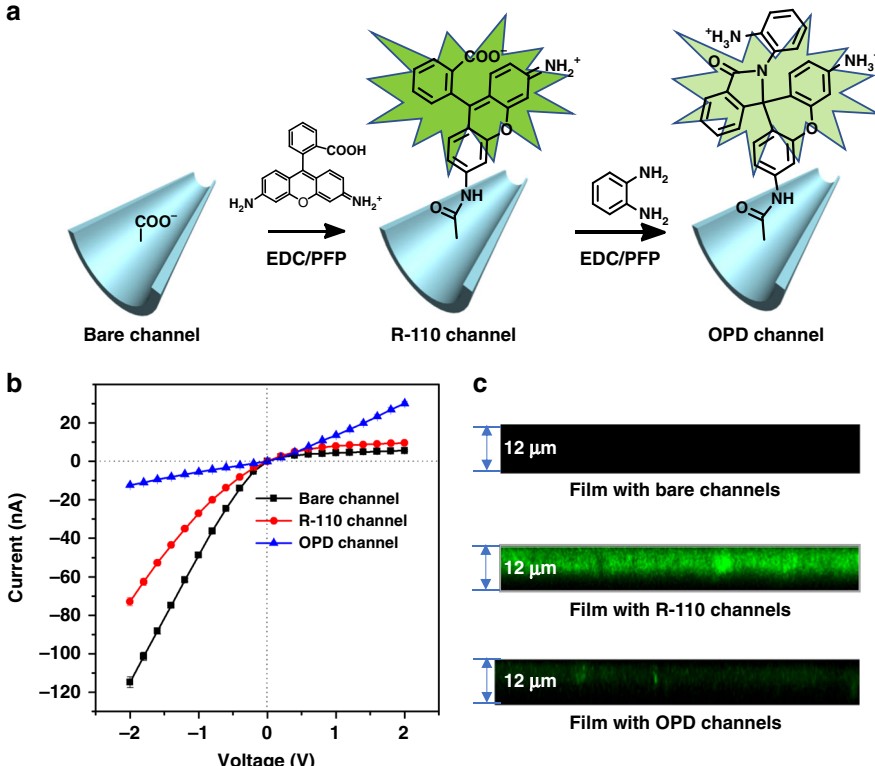

**Fig. 2** Fabrication of the NO-regulated nanochannel. **a** Process for the construction of the NO-gated nanochannel by using a spiro ring opening/closing reaction strategy; **b** $I-V$ properties of the single nanochannel before and after modification in 0.1 M KCl; **c** LSCM observation of the fluorescent signals before and after R-110 and OPD modification. LSCM laser-scanning confocal microscopy

$59.0 \pm 1.5°$ to $72.2 \pm 1.8°$ and $82.1 \pm 2.2°$, respectively), corresponding to a change in the chemical composition. X-ray analysis shows a new $N$ peak at 400 eV after R-110 immobilization and a significant increase in the $N$ peak after subsequent OPD conjugation (Supplementary Fig. 3). The progress of R-110 and OPD modification of the nanochannel surface was tracked by laser-scanning confocal microscopy (LSCM). As shown in Fig. 2c, a strong fluorescence signal appears after R-110 modification. However, conjugation of OPD with R-110 leads to spirocyclic formation, resulting in a dramatic decrease in the fluorescence signal.

**The ionic rectification and gating ratio**. After successful construction of the nanofluidic diode, we investigated the feasibility of the nanochannel in response to NO. When NO was bubbled (~2 mM) into this system, the amino groups of the grafted OPD molecules reacted with NO, inducing an opening of the spiro rings and release of OPD derivatives into the solution. Therefore, the surface chemical composition recovered from OPD molecules to R-110 molecules, and the surface recovered its negative charge owing to exposure of the –COOH groups of R-110, leading to current increase from −12.3 to −60.9 nA at −2 V and immediate reversal of the rectifying characteristics (Fig. 3a). In contrast, the $I-V$ curves of the bare and R-110-modified nanochannels show almost no change after bubbling of NO (Supplementary Fig. 4), which confirms the specific interaction between NO and OPD molecules. The reversal of ion rectification was attributed to the corresponding change of surface charge in the conical nanochannel. The ion rectification ratio was calculated by the absolute values of the ion currents at a given voltage −2 V versus +2 V. The calculated rectification ratio is only 0.4 for the OPD-modified nanochannel and it reaches 6.7 after bubbling of NO (Supplementary Fig. 5).

Similar to ion channels in cells, we also investigated whether the gating state of the nanodevice can be efficiently regulated by NO. The influence of NO on ion transport through the channel in terms of the gating ratio is defined as $Rg = (I-I_0)/I_0$, where $I_0$ and $I$ are the currents measured at −2 V before and after treatment with NO. As shown in Fig. 3b, the gating ratio is 4.0 for the OPD-modified nanochannel (ON state), which is significantly higher than the values for the bare and R-110-modified nanochannels (OFF state). Moreover, recovery of the fluorescence signals in the nanochannel in the presence of NO indicates that NO efficiently induces an opening of the spiro rings (Fig. 3c). Furthermore, the CA changes from $(82.1 \pm 2.2°)$ to $(72.9 \pm 1.8°)$ after bubbling. NO reflects the wettability change of the channel surface (Fig. 3d). The increase in the hydrophilicity is because of the removal of OPD molecules from the channel surface. Finally, we investigated the reversibility of the NO-responsive nanochannel. The cycling performance of the system was investigated by recording the current change at −2 V between the OFF states and ON states of the nanochannel (Fig. 3e). Further, the $I-V$ curve, contact angle, and fluorescence confocal microscopy also confirmed this process (Supplementary Figs. 6–8). When the nanochannel is activated with NO to open the spiro rings, the nanofluidic diode is in the ON state with a high current level, which is attributed to exposure of the –COOH groups of R-110 and the remarkable increase in the negative charge. However, when the OPD molecule again binds to the nanochannel, the nanochannel is in the OFF state with a low current level at −2 V. After seven cycles, no damping of the ion current is observed, indicating that the ionic gate has excellent stability and repeatability performance to regulate ionic conduction of the nanochannel.

**The selectivity, responsive rate, and switchability**. To verify the high selectivity of the NO-regulated ion gate, the OPD-modified

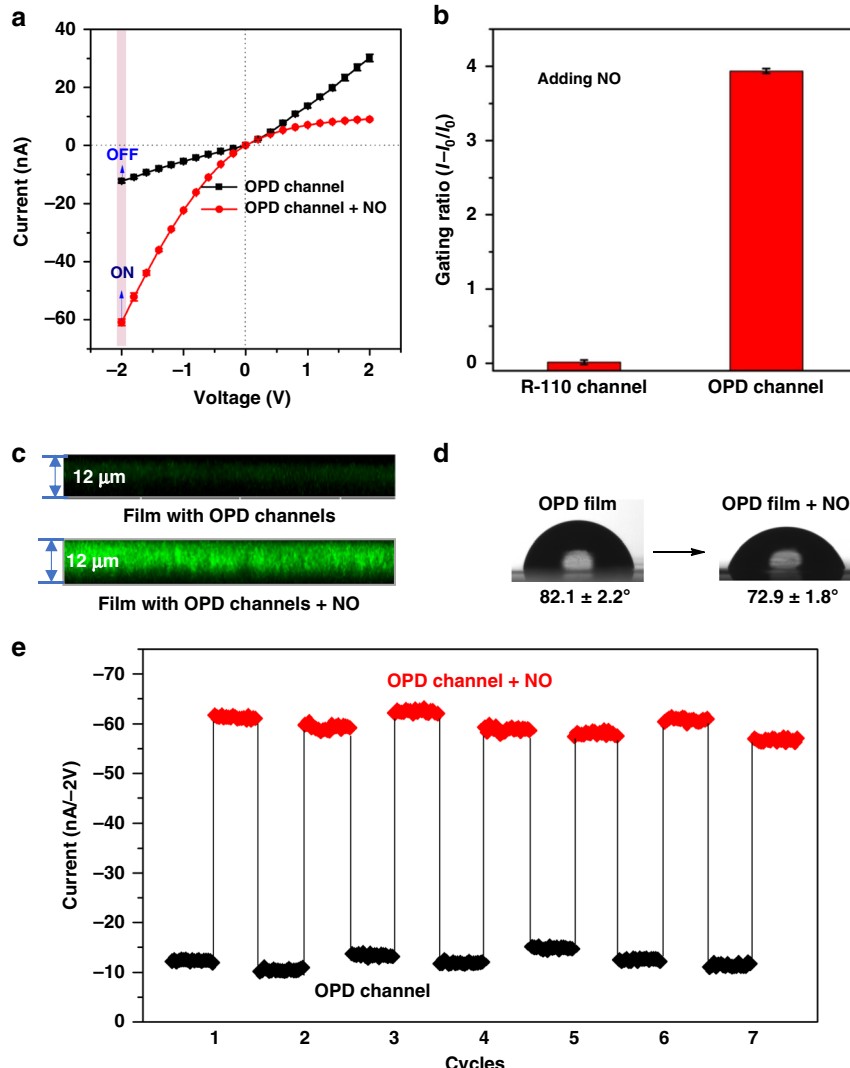

**Fig. 3** NO-driven ionic rectification and gate. **a** $I-V$ responses of the OPD channel in the absence and presence of NO; **b** NO-driven ion gating ratio (Rg), which shows that the OPD channel was opened by NO; **c** LSCM observation of fluorescence signals of the OPD-modified nanochannel in the absence and presence of NO; **d** photographs of the water droplet shape on the PET films in the absence and presence of NO; **e** stability and switchability of the NO-responsive ionic gate. OPD o-phenylenediamine, LSCM laser-scanning confocal microscopy, PET polyethylene terephthalate

nanochannel was tested with different types of gases, ions, and molecules for 3 h. As shown in Fig. 4a, the ionic current at −2.0 V increases from −5.5 to −26.5 nA after the gate was activated by NO. However, the ion current remains at about −5.0 nA in the presence of other substrates, which indicates that the OPD-modified nanochannel shows a high selectivity for NO. The reason could be explained that only NO specifically attacks the spiro rings and exposes the −COOH groups of R-110. The gating ratio of NO (~3.8) is much higher than those of the other substrates, confirming the selectivity for NO attack. The ion transport properties of this system were also investigated by current measurements at different NO concentrations. The ion current and gating ratio obviously increased with the enhancement of NO concentration (Supplementary Fig. 9). Moreover, the nanochannel could still respond to NO even down to 0.1 μM. The $I-V$ curve can reach a platform when the concentration of NO is above 0.1 mM. To evaluate the NO response rate and switchability of the ion channel, we recorded the current with increasing NO ventilation time. As NO is continuously bubbled, the ion current gradually increases, which indicates that the nanochannel gradually opens (Fig. 4b, Supplementary Fig. 10). When the

bubbling time is >140 min, the ion current no longer increases owing to saturation of the reaction between NO and OPD. The nanochannel is then re-modified with OPD molecules, and the gating of the system returns to the OFF state. After repeating the process seven times, no damping of the ion current is observed, indicating that this strategy guarantees the NO-driven ionic gate with excellent switchability and stable reversibility.

**The mechanism of the NO-responsive nanochannel.** Due to the reversal of current rectification, the reason for the ionic gate transition between the ON and OFF states is considered to be the polarity conversion of the surface charge regulated by NO (Fig. 5a). Therefore, the charge densities of the R-110 channel and the OPD channel were first tested by Probstein's classic text and then calculated by the Helmholtz−Smoluchowski equation (see ESI and Supplementary Table 5)[17,48–50]. The result of the electroosmotic flow (EOF) calculation indicated that the OPD channel is positive charged (charge density: +0.05 e nm$^{-2}$) and the R-110 channel is negative charged (charge density: −0.13 e nm$^{-2}$). Moreover, the change of surface charge density of the OPD

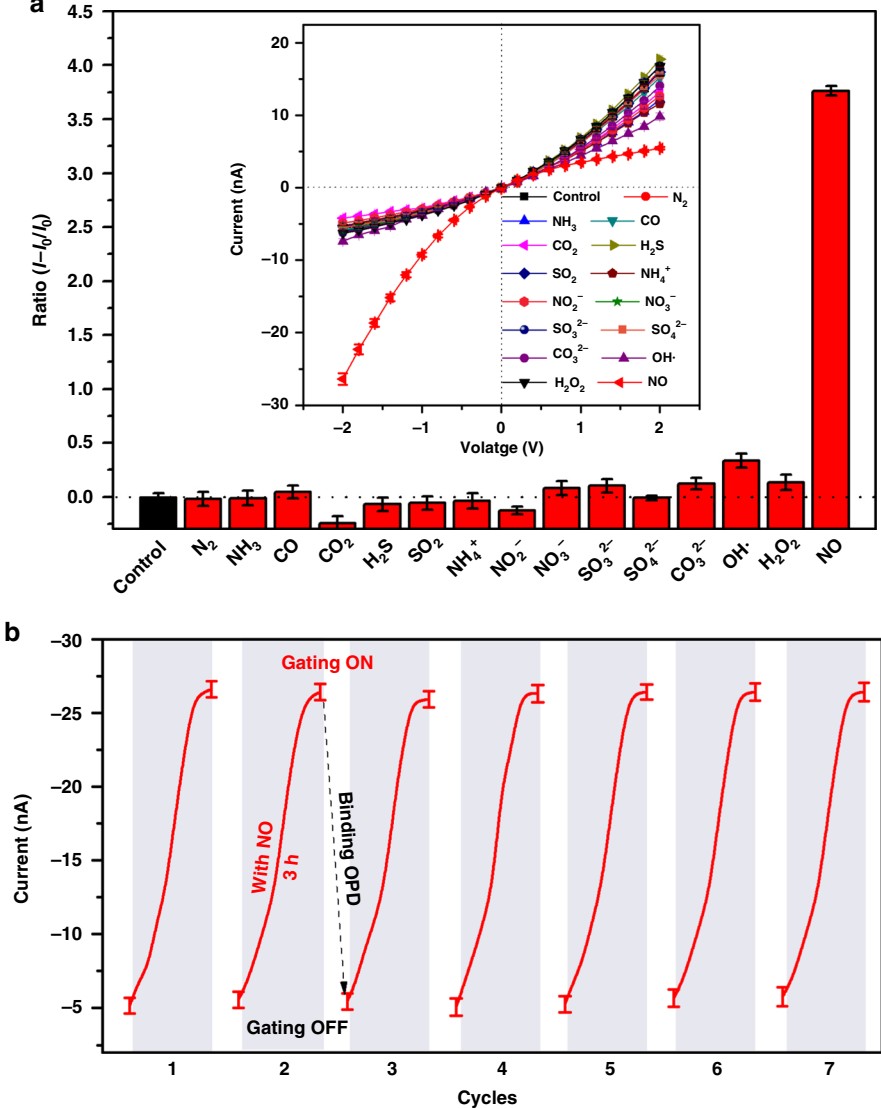

**Fig. 4** Characterization of the gating performance. **a** Gating ratios measured at −2 V in 0.1 M KCl, with addition of 1 mM various substrates; **b** the dynamic changes and recyclability of the NO-responsive nanochannel

channel is dynamic from +0.05 e nm$^{-2}$ to −0.13 e nm$^{-2}$ with the increase of NO concentration as shown in Fig. 5a.

Furthermore, the significant transformation of surface charge is attributed to the change of the number/ratio of the exposed −NH$_2$ groups and −COOH groups due to the spiro ring opening/closing. The electrospray ionization–mass spectrometry (ESI–MS) analysis, the strong fluorescence and color change, and the $^1$HNMR confirmed the NO-regulated process involving cyclization/opening of the spiro ring in solution (Supplementary Figs. 11–13). When the spiro ring closes, OPD molecules with more −NH$_2$ groups coat the channel surface, and the protonation of the −NH$_2$ at a neutral pH results in the net positive charge on the channel surface. While NO induces the spiro ring opening, the exposure of deprotonated −COOH on R-110 results in the net negative charge on the channel surface. As numerous reports identified, the EOF in the negatively charged conical nanochannel is prone to transport from the tip to the base, resulting in rectified ion current ($R = I_{−2 V}/I_{+2 V} > 1$) as the ON state of the diode. The decrease of charge density switches the diode to the OFF state, as NO regulates the negatively charged R-110 channel ($R = 6.7 > 1$) to a slight positively charged OPD channel ($R = 0.4 < 1$). The dynamically changed ionic current with an increasing

concentration of NO (Supplementary Fig. 9) also demonstrates the relationship between the ionic gate switch and the surface charge reversal. Therefore, the result of polarity conversion of surface charge is the ionic gate switch between the ON and OFF states.

To further prove the gating mechanism that the NO-regulated surface charge reversal switches ionic gating, finite-element computations based on the Poisson and Nernst−Planck equations by COMSOL Multiphysics 5.3 were performed to demonstrate the behavior of ions in the two extreme channels with the corresponding charge density (for details, see the Methods section and Supplementary Fig. 14)[51,52]. Figure 5b shows the two-dimensional profile of ion concentration at −2 V. It is obvious that a rather larger concentration than the bulk solution appeared at a negatively charged channel (simulation of the R-110 channel) as ion enrichment. Conversely, ion depletion takes place in a positively charged channel (simulation of the OPD channel). The ion enrichment owes to the overlap of the electric double layer at the tip side of the negatively charged channel, and results in the forward concentration gradient generating a higher ion conductance as well as high ionic current. Therefore, as a summary of the mechanism, the NO-regulated

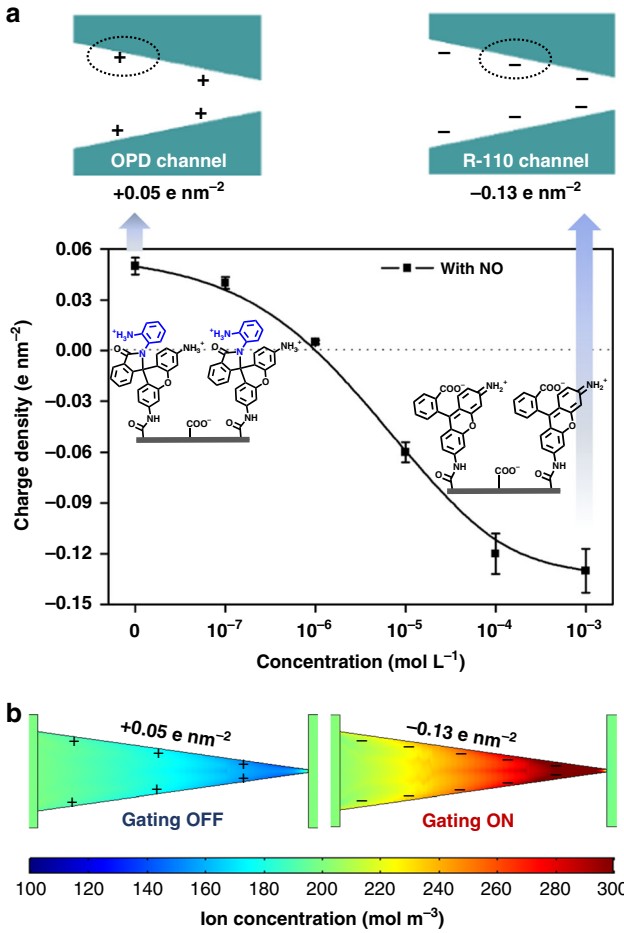

**Fig. 5** Discussion of the mechanism. **a** The surface charge density of the OPD channel and the R-110 channel, and the dynamic changes of surface charge density from the OPD channel to the R-110 channel (±10% was used for describing the error bars). **b** The numerical simulation of ion concentration (total of K$^+$ and Cl$^-$) distribution in the channel (at −2 V, with 0.1 M KCl). OPD o-phenylenediamine

chemical structure change in the channel leads to polarity conversion of the surface charge from positive to negative, and the negatively charged R-110 channel generates a higher ion conductance and a greater ion transmission as an ON state of the ionic gating.

## Discussion

In summary, we have successfully fabricated an NO-regulated artificial ion nanochannel with excellent gating and rectifying properties. The switchability of the NO-regulated nanochannel is attributed to the spiro ring opening−closing chemical strategy, which leads to a significant change of surface charge. Driven by a −2 V potential, the ion current is transported from the tip end to the base end. When the spiro ring forms (closing), the charge of the channel surface becomes positive (+0.05 e nm$^{-2}$), resulting in inhibition of the ion transportation activity (OFF state). However, the ion transportation ability is triggered (ON state) by NO-induced spiro ring opening due to the presence of a negative charge (−0.13 e nm$^{-2}$) on the nanochannel surface. Therefore, this nanofluidic diode exhibits the macroscopic ion gating behavior under external NO. Moreover, this NO gating system exhibits excellent reversibility and stability, as well as the high selectivity, which will help to better understand the signal function of NO in the ion channels of living organisms. Considering

the high selectivity and reversibility for NO, this NO-responsive nanochannel will have potential applications in NO biosensors.

## Methods

**Nanochannel construction**. In order to create a single conical nanochannel, the polyethylene terephthalate membranes (treated with the ion track-etching technique) were irradiated under ultraviolet light condition for 60 min (Supplementary Fig. 15). Generally, the first chemical etching should be performed from one side with sodium hydroxide (concentration: 9 M) and the other side with potassium chloride and formic acid (concentration: 1 M) as stopping solution. During this experiment, scanning voltage (1 V) was used to trace (20 min at 35 °C). When the current value reached 2.5 nA, potassium chloride and formic acid (concentration: 1 M) were added into the NaOH-treated side for 20 min. After that, the second chemical etching should be performed from both sides using sodium hydroxide (concentration: 1 M). Once the transmembrane current arrives at about 50 nA (about 2 h), the stopping solution potassium chloride and formic acid (concentration: 1 M) was immediately added to neutralize the NaOH. Diameter calculation of the base (D) was verified by SEM (about 700 nm) and the nanochannel tip estimated from the equation is about 23 nm.

**OPD and R-110 immobilization on the nanochannel**. The free carboxyl groups of the as-prepared nanochannel walls were immobilized with R-110 via a traditional coupling strategy [1-(3-dimethylaminopropyl)-3-ethylcarbodiimidehydrochloride and pentafluorophenol (EDCI/PFP) reagents, 10 mM EDCI/PFP solution for 1 h]. Moreover, the R-110-modified nanochannel was further conjugated with OPD molecule (10 mM R-110 ethanol solution overnight and rinsed with deionized H$_2$O several times).

**Current measurement**. A Keithley 6487 picoammeter was used to perform the survey of currents. Transmembrane potential across the nanochannel was identified by Ag/AgCl electrodes (Supplementary Fig. 15). The I−V curves of this work were collected via the varied scanning voltage (−2 to +2 V, 20-s period). The average current values at different voltages were collected through five repeated tests.

**Contact angle measurement**. Generally, the collection of CAs was achieved via an OCA 20 system. To collect the CA data of the bare membrane, sodium hydroxide (concentration: 9 M) as the chemical etching solution treated the PET membrane. After 60 min, potassium chloride and formic acid (concentration: 1 M) as a stopping solution treated the membrane for 30 min and then rinsed it with deionized H$_2$O. The average CA value was achieved through five different locations. The contact angle of the OPD film and the R-110 film was measured by the same treatment method.

**X-ray photoelectron spectra experiments**. The collection of XPS data was carried out by an ESCALab220i-XL electron spectrometer. The new N peak at 400 eV verified the successful modification of R-110 and OPD.

**Finite-element computation**. Finite-element computations were performed on the Poisson and Nernst−Planck equations using COMSOL Multiphysics 5.3. The equations are shown below:

$$J_i = D_i \left( \nabla c_i + \frac{z_i F c_i}{RT} \nabla \varphi \right) + u c_i, \tag{1}$$

$$\nabla^2 \varphi = -\frac{F}{\varepsilon} \sum z_i c_i, \tag{2}$$

$$\nabla \cdot J_i = 0. \tag{3}$$

Equation 1 is the Nernst–Planck equation that describes the transport property of a charged nanochannel. The electric potential and ionic concentration can be characterized by Eq. 2. Besides, the flux should satisfy the time-independent continuity Eq. 3 when the system reaches a stationary regime. The physical quantities $J_i$, $D_i$, $c_i$, $\varphi$, $u$, $R$, $F$, $T$, and $\varepsilon$ refer to the ionic flux, diffusion coefficient, ion concentration, electrical potential, fluid velocity, universal gas constant, Faraday constant, absolute temperature, and dielectric constant of the electrolyte solutions, respectively. The coupled equations can be solved by assuming appropriate boundary conditions. The boundary condition for potential $\varphi$ on the channel wall is Eq. 4; and the ionic flux has zero normal components at boundaries (Eq. 5), where $\sigma$ represents the surface charge density, and is given by the EOF experiment.

$$\overrightarrow{n} \cdot \nabla \varphi = -\frac{\sigma}{\varepsilon}, \tag{4}$$

$$\overrightarrow{n} \cdot J_i = 0. \tag{5}$$

Electrolyte solution was 0.1 M KCl, and the diffusion coefficients of K$^+$ and

$Cl^-$ were set as $2.0\times10^{-9}\,m^2\,s^{-1}$. The simulated model is shown in Supplementary Fig. 14.

## Data availability

The authors declare that the data supporting the findings of this study are available within the article and its Supplementary Information files, and all data are available from the authors on reasonable request.

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

## Acknowledgements

This work was financially supported by the National Key Research and Development Program of China (2018YFD0200102, 2017YFA0505203), the National Natural Science Foundation of China (21572076, 21772055, and 21708012), the Nature Science Foundation of Hubei Province (2018CFB534), the 111 Project (B17019), Wuhan scientific and technological projects (2015020101010079), and the Open Research Fund of State Key Laboratory of Chemo/Biosensing and Chemometrics (2017002).

## Author contributions

Y.S., G.Y. and H.L. conceived and designed the experiments. Y.S. and S.C. made a major contribution in all experiments. Y.S., S.C., X.C., Y.X. and H.L. wrote the paper. S.Z., Q.O., G.Y. and H.L. supervised all experiments and analyses.

## Additional information

**Competing interests:** The authors declare no competing interests.

