## [Peer Review File · Nature Communications]

Reviewers' comments:

Reviewer #1 (Remarks to the Author):

In this manuscript, a type of solid phase channels with gating behavior was described. The gating of the channel was caused by a NO-regulated spiro ring opening/closing reaction. Here are my suggestions for revision:

1. The word "bioinspired" is unsuitable for this work, because the channel does not contain any biological elements.
2. Why did the microscopic change (ring opening/closing) in the channel structure lead to the macroscopic gating behavior? The mechanism should be further discussed in the text.
3. It was proposed that the positively charged channel was in OFF state due to its blockage to K^+ . In principle, the positive charges should facilitate the flux of Cl^- . Thus, the negative current could be observed. However, we could not observe it. Why?

Reviewer #2 (Remarks to the Author):

The manuscript entitled "A bioinspired NO-regulated K^+ gate based on a spiro ring opening/closing reaction strategy" reports a biomimetic NO-regulated K^+ gate based on a spiro ring opening/closing reaction mechanism. The nanochannel system exhibits an outstanding NO selectivity, good reversibility and stability, owing to the specific reaction between o-phenylenediamine (OPD) and NO on the channel surface. The nanofluidic diode shows a high K^+ ion rectification ratio and ion gating ratio. It is believed that the finding in this paper would be of help in understanding the synergistic effect of NO messengers in the ion channels of living organisms. However, the authors should consider the following points in any form of revision:

1. The author claims that the bioinspired NO-regulated K^+ gate shows excellent reversibility and stability. However, the cycles are too few to support this point.
2. The author uses the same figure in Fig. 1b and Fig. 5a, which is not reasonable.
3. What is the feeding amount of NO and what is the concentration of o-phenylenediamine? The authors should incorporate this information in the manuscript.
4. From Fig. 2c, it seems that there are still some fluorescent signals in the films with OPD channels, therefore, the repeatability would be an issue. How is the result after many cycles, say, 10 cycles.
5. What is the lowest concentration to achieve the permeation of K^+ ?
6. The authors show the ESI-MS analysis to explain the mechanism of NO-driven ionic gate, however, the authors also need to show the HNMR results before and after dealing with NO (page 4).
7. The authors clearly indicate the process of the NO-gated nanochannel, to better understand this article for the readers, please also show the detailed ring opening/closing mechanism step by step from R-100 channel to OPD channel (page 3), not just the products before and after reactions, which also can provide a clear proof for Fig. 4A.
8. The research has a lack of systematic work. The only type of data are Current Curves. The authors need to provide more kinds of means, such as dynamic simulation, to support the points.
9. Is the shape and size repeatable during every fabrication?
10. Citation of this paper is informative. But many recent papers related to nanopore and nanochannels have been published, such as, Nano Today 2018, 20: 84-100, ACS Nano 2018, 12(2): 908-911, Advanced Materials 2016, 28: 7049-7064, Advanced Functional Materials 2015, 25(7): 1102-1110, ACS Nano 2015, 9(12): 122264-12273, ACS Applied Materials & Interfaces 2013, 5(16): 7931-7936, Small 2014, 10(4): 793-801, Chemical Communications 2012, 48(47): 5901-5903, Chemical Communications 2011, 47 (11) :3102-3104, Advanced Materials 2010, 22(22): 2440-2443, et al. In order to help readers better understand the importance of this work, these references should be considered to cite accordingly, correspondingly.

I would recommend a revised version of this manuscript for publication in Nature Communications.

Reviewer #3 (Remarks to the Author):

The authors demonstrated a bioinspired NO-regulated K⁺ gate based on a spiro ring opening/closing reaction strategy. This nanofluidic diode system shows NO selective response, and the authors declare that the system exhibits fast response, excellent reversibility and stability. However, there are some questions needed to be addressed.

1. The detailed response mechanism of K⁺ channel in smooth muscle cells should be provide in figure 1a. Figure 1b shows that the biomimetic NO-driven ion gate response to both NO and O₂, but Figure 4a shows this system does not response to O₂, it is really confusing, the authors should explain.

2. There are many responsive nanochannels to regulate K⁺ have been reported before, which should be discussed and compared with this work in the introduction section.

3. The authors say this NO gating system exhibits excellent reversibility and stability as well as fast response rate. But, I think to claim "excellent", more cycles need to be shown. Figure 4b, shows NO is continuously bubbled, current gradually increasing, which indicates that nanochannel gradually opens. And it takes 140 min to reach a stable current which could not call fast response (but it shows 140 s in the main text, really confusing! the authors should be more careful to their results)

4. The authors call the system as a bioinspired NO-regulated K⁺ gate which is not accurate. As shown in figure 1 and figure 5, this system has no specific choice for potassium ions but also can response for all cations.

5. I don't think the Numerical simulations of the NO-regulated nanochannel make any sense. And the corresponding formulas and models used in the simulations are also unclear.

Reviewers' comments:

Reviewer #1 (Remarks to the Author):

In this manuscript, a type of solid phase channels with gating behavior was described. The gating of the channel was caused by a NO-regulated spiro ring opening/closing reaction. Here are my suggestions for revision:

Response:

First of all, we sincerely thank the Referee 1 for her/his critical reading of our manuscript, which help to significantly improve our manuscript.

1. The word “bioinspired” is unsuitable for this work, because the channel does not contain any biological elements.

Response:

Thanks for pointing out this issue. Following your suggestions, we reconstructed a new title “A highly selective and recyclable NO-responsive nanochannel based on a spiro ring opening/closing reaction strategy” in the revised manuscript.

2. Why did the microscopic change (ring opening/closing) in the channel structure lead to the macroscopic gating behavior? The mechanism should be further discusses in the text.

Response:

Thanks for pointing out this issue. Following your suggestion, we have added more detail information on the sections of “The mechanism of NO-regulated nanochannel” and “Discussion” in the revised manuscript. In order to better understand the mechanism, we also describe it as the following figures:

3. It was proposed that the positively charged channel was in OFF state due to its blockage to K^+ . In principle, the positive charges should facilitate the flux of Cl^- . Thus, the negative current could be observed. However, we could not observe it. Why?

Response:

As previous reports identified, the electroosmotic flow in negative charged conical nanochannel drives K^+ more prone to transport from the tip to the base (under experimental condition of applied -2V potential), resulting into rectified ion current ($R=I_{-2V}/I_{+2V}=7.6>1$).¹⁻³ The electroosmotic flow in positive charged conical nanochannel drives Cl^- more prone to transport from the tip to the base (under experimental condition of applied +2V potential), resulting into the opposite rectification ($R=I_{-2V}/I_{+2V}=0.4<1$). Additionally, the extent of ion current and rectification is influenced by surface charge density. The higher charge density, the higher ion current and rectification. In our work, the charge density of positive charged OPD channel is slight, therefore the negative current as you mentioned is not obvious.

(In the revised article, we did not focus on the phenomenon of K^+ transmission, but further studied the mechanism of NO-regulated ion current.)

1. Ali, M. et al. A pH-tunable nanofluidic diode with a broad range of rectifying properties. *ACS Nano*. 3, 603–608 (2009).
2. White, H. S. & Bund, A. Ion Current Rectification at Nanopores in Glass Membranes. *Langmuir*. 24, 2212–2218 (2008)
3. Xiao, K. et al. Enhanced Stability and Controllability of an Ionic Diode Based on Funnel-Shaped Nanochannels with an Extended Critical Region. *Adv. Mater.* 28, 3345–3350 (2016).

Reviewer #2 (Remarks to the Author):

The manuscript entitled “A bioinspired NO-regulated K^+ gate based on a spiro ring opening/closing reaction strategy” report a biomimetic NO-regulated K^+ gate based on a spiro ring opening/closing reaction mechanism. The nanochannel system exhibits an outstanding NO selectivity, good reversibility and stability, owing to the specific reaction between o-phenylenediamine (OPD) and NO on the channel surface. The nanofluidic diode shows a high K^+ ion rectification ratio and ion gating ratio. It is believed that the finding in this paper would be of help in understanding the synergistic effect of NO messengers in the ion channels of living organisms. However, the authors should consider the following points in any form of revision:

Response:

First of all, we sincerely thank the Referee 2 for her/his critical reading of our manuscript, which help to significantly improve our manuscript.

1. The author claims that the bioinspired NO-regulated K^+ gate show excellent reversibility and stability. However the cycles are too few to support this point.

Response:

Thanks for pointing out this issue. Following your suggestions, we have done more

cycle experiments in the revised manuscript.

Fig. 3e Stability and switchability of the NO-responsive ionic gate.

2. The author use the same figure in Fig.1b and Fig.5a, which is not reasonable.

Response:

We are very sorry for the confusion caused by the Fig.1b and Fig.5a. We have reconstructed Fig.1b and Fig.5a in the revised manuscript.

3. What is the feeding amount of NO and what is the concentration of o-phenylenediamine? The authors should incorporate these information in the manuscript.

Response:

Thanks for pointing out this issue. Following your suggestions, we have added these detail information in the revised manuscript as following:

By bubbling, the NO concentration is about 2 mM. (at room temperature, the fresh saturated NO concentration is 2 mM) and the concentration of o-phenylenediamine is 10 mM.

4. From Fig.2c, it seems that there are still some fluorescent signals in the films with OPD channels, therefore the repeatability would be an issue. How is the result after many cycles, say, 10 cycles.

Response:

Thanks for pointing out this issue. Following your suggestions, we have carried out the cycle experiment for 10 times and added the fluorescent signal comparison data in the revised supported information (Supplementary Fig. 7). Moreover, we also verified

the change of contact angle for 10 cycles (Supplementary Fig. 6) in the revised supported information.

5. What is the lowest concentration to achieve the permeation of K^+ ?

Response:

Thanks for pointing out this issue. According to the previous literature method^{1,2}, in this work, the concentration of NO achieve the permeation of K^+ down to 0.1 μ M. (supplementary Fig.9).

Reference:

1. J. Wang, J. Hou, H. Zhang, Y. Tian, L. Jiang. *ACS. Appl. Mater. Interfaces*. **2018**, 10, 2033.
2. B. Niu, K. Xiao, X. Huang, Z. Zhang, X. Kong, Z. Wang, L. Wen, L. Jiang. *ACS. Appl. Mater. Interfaces*. **2018**, 10, 22632.

6. The authors show the ESI-MS analysis to explain the mechanism of NO-driven ionic gate, however the authors also need to show the HNMR results before and after dealing with NO (page 4).

Response:

Thanks for your good suggestion. Following your suggestion, we have added the ^1H NMR results of ring opening/closing in the revised supported information (supplementary Fig.13).

7. The authors clearly indicate the process the NO-gated nanochannel, to better understand this article for the readers, please also show the detailed ring opening/closing mechanism step by step form R-100 channel to OPD channel (page 3), not just the products before and after reactions, which also can provide a clear proof for Fig. 4A.

Response:

Thanks for your good suggestion. Following your suggestion, we have added the detail information on the ring opening/closing mechanism step from R-100 channel to OPD channel(supplementary Fig.11) in the revised manuscript as following:

8. The research has a lack of systematic work. The only type of data are Current Curves. The authors need to provide more kinds of means, such as dynamic simulation, to support the points.

Response:

Thanks for pointing out this issue. Following your suggestion, We added dynamic numerical calculations of surface charges in the revised manuscript and supported

information (Fig 5 a. and Supplementary Table 5).

9. Do the shape and size repeatable during every fabrication?

Response:

Ion track technology is widely reported, tapered nanochannel is a common and easily prepared shape. The shape and size can be reproduced by strictly controlling the etching conditions (etching solution concentration, temperature and etching time), at the same time, transmembrane current is used as a reference. Our preparation conditions are added in section of “Nanochannel Preparation” in the revised manuscript.

10. Citation of this paper is informative. But many recent papers related to nanopore and nanochannels have been published, such as, Nano Today 2018, 20: 84-100, ACS Nano 2018, 12(2): 908-911, Advanced Materials 2016, 28: 7049-7064, Advanced Functional Materials 2015, 25(7): 1102-1110, ACS Nano 2015, 9(12): 122264-12273, ACS Applied Materials & Interfaces 2013, 5(16): 7931-7936, Small 2014, 10(4): 793-801, Chemical Communications 2012, 48(47): 5901-5903, Chemical Communications 2011, 47 (11) :3102-3104, Advanced Materials 2010, 22(22): 2440-2443, et al. In order to help readers better understand the importance of this work, these references should be considered to cite accordingly, correspondingly.

Response:

Thanks for your good suggestion. We have added these papers in the revised manuscript.

I would recommend a revised version of this manuscript for publication in Nature Communications.

Reviewer #3 (Remarks to the Author):

The authors demonstrated a bioinspired NO-regulated K⁺ gate based on a spiro ring opening/closing reaction strategy. This nanofluidic diode system shows NO selective response, and the authors declare that the system exhibits fast response, excellent reversibility and stability. However, there are some questions needed to be addressed.

Response:

First of all, we sincerely thank the Referee 3 for her/his critical reading of our manuscript, which help to significantly improve our manuscript.

1. The detailed response mechanism of K⁺ channel in smooth muscle cells should be provide in figure 1a. Figure 1b shows that the biomimetic NO-driven ion gate response to both NO and O₂, but Figure 4a shows this system does not response to O₂, it is really confusing, the authors should explain.

Response:

Thanks for pointing out this issue.

Figure 1a shows a cartoon picture about NO directly activate single ion channel. In *vivo*, NO is generally thought to indirect activate K⁺ channel by stimulate the guanylate cyclase in smooth muscle cell. Even though some studies suggest that NO can directly activate single K⁺ channel without guanylate cyclase, of which the mechanism is still unclear. (**References 6** *Bolotina, V. M. et al. Nitric oxide directly activates calcium-dependent potassium channels in vascular smooth muscle, Nature, 368, 850–853 (1994).*) Therefore, inspired by the latter, the cartoon picture in figure 1a is imitating the directly NO-activated ion channel, and the strategy in figure 1b is designed to establish an artificial ion channel with the function of direct NO-activation.

We are sorry to make this confusion. In this study, O₂ is from the air. O₂ is necessary for the cyclization reaction to carry out, but the separate O₂ cannot work (as shown in figure 4a), therefore this system is considered to response to NO. In order to avoid the confusion, we have changed the O₂ to air in Figure 1b and removed the O₂ in Figure

4a in the revised manuscript.

2. There are many responsive nanochannels to regulate K^+ have been reported before, which should be discussed and compared with this work in the introduction section.

Response:

Thanks for pointing out this issue. Following your suggestion, we have added more discussion on the comparison between previous responsive nanochannel and this work in the introduction section of the revised manuscript as following:

Most of previous report responsive nanochannels to regulate K^+ are based on the non-covalent binding such as electrostatic interactions, van der Waals forces, and hydrogen bonds between the external stimuli and ligand on the nanochannel surface. For example, Jiang et al. reported an functional nanochannel to regulate K^+ based on the conformational change of the G-quadruplex DNA. Azzaroni et al. recently reported an pH-responsive nanochannel to regulate K^+ based on the polyelectrolytes self-assembly inside the channel surface. However, these non-covalent interactions may result in subtle interference with ligands on the channel surface and decrease the selectivity of channel regulation, especially for gas-driven systems. Furthermore, most of these systems could not achieve a selective response in a complex matrix and for real samples. Recently, our group reported a thiol-yne reaction-based nanochannel responsive system that exhibits high cysteine specificity and selectivity even in complex matrices.³⁹ Despite this success, the thiol-yne strategy is essentially irreversible, resulting in recycling of this responsive system being impossible.⁴⁰⁻⁴¹ Therefore, developing a universal strategy for constructing a gas-driven nanochannel with high selectivity and recyclability is urgently required.

3. The authors say this NO gating system exhibits excellent reversibility and stability as well as fast response rate. But, I think to claim "excellent", more cycles need to be shown. Figure 4b, shows NO is continuously bubbled, current gradually increasing, which indicates that nanochannel gradually opens. And it takes 140 min to reach a stable current which could not call fast response (but it shows 140 s in the main text,

really confusing! the authors should be more careful to their results)

Response:

Thanks for pointing out this issue. Following your suggestion, we have done more cycles to demonstrate the good reversibility and stability in the revised manuscript.

Moreover, we are very sorry to make this mistake. We have corrected the 140s to 140 min and delete the fast response in the revised manuscript.

4. The authors call the system as a bioinspired NO-regulated K^+ gate which is not accurate. As shown in figure 1 and figure 5, this system has no specific choice for potassium ions but also can response for all cations.

Response:

Thanks for pointing out this issue. In this work, the selectivity of K^+ isn't discussed in the goal and design of this work. Here the KCl is used as a general ionic electrolyte to show the Gating phenomenon in this nanochannel.

Following your suggestions, we have reconstructed the title to "A highly selective and recyclable NO-responsive nanochannel based on a spiro ring opening/closing reaction strategy" in the revised manuscript.

5. I don't think the Numerical simulations of the NO-regulated nanochannel make any sense. And the corresponding formulas and models used in the simulations are also unclear.

Response:

Thanks for pointing out this issue. We are very sorry for this unclear description.

In the revised manuscript and supporting information, we provide the models,

formulas, and corresponding parameter setting of the numerical simulation. Firstly, the finite-element computations is processed by COMSOL Multiphysics 5.3 based on the Poisson and Nernst-Planck equations. Then the geometry of model conforms with the shape and size of nanochannel, the parameters of electrolyte and voltage are same as the experiment condition. Furthermore, as our hypothesis that the ion is gating switch results from surface charge reversal, we set the positive charge densities to the OPD channel ($+0.05e^-/nm^2$) and R-110 channel ($-0.13e^-/nm^2$), which is tested by the EOF experiment and calculated by Helmholtz-Smoluchowski equation.

Concentration profile show that rather larger concentration than bulk solution appeared at negative charged channel, (simulation of R-110 channel), as ion enrichment. The ion enrichment at the tip results to low ion resistance, therefore the ionic gating ON and generates high ionic current as well as the rectified ion current ($R=I_{-2V}/I_{+2V}>1$). Conversely, ion depletion in slightly positive charged channel (simulation of OPD channel) results to gating OFF and generates low current and reversed rectification. Therefore, as a summary of the mechanism, the NO-regulated chemical structure change in the channel leads to polarity conversion of the surface charge from positive to negative, and the negative charged R-110 channel generates higher ion conductance and greater ion transmission as a ON state of the ionic gating. The numerical simulation rigorously agrees with the experiment results and verifies our mechanism.

The corresponding formulas and models we have replenished in the Methods, Supplementary Methods and Supplementary Fig. 14, Table 5.

Reference:

1. White, H. S. & Bund, A. Ion Current Rectification at Nanopores in Glass Membranes. *Langmuir*. 24, 2212–2218 (2008).
2. Xiao, K. et al. Enhanced Stability and Controllability of an Ionic Diode Based on Funnel-Shaped Nanochannels with an Extended Critical Region. *Adv. Mater.*28, 3345–3350 (2016).

REVIEWERS' COMMENTS:

Reviewer #1 (Remarks to the Author):

The manuscript has been well revised according to the reviewers' comments and suggestions. I agree its publication as it stands.

Reviewer #2 (Remarks to the Author):

The paper is well explained. Most of the previous comments were addressed in the revised manuscript. Some minor comments should be considered:

1. The label fonts in the figures need to be unified.
2. There are still some grammatical errors in the article that need to be further carefully modified.
3. Please show each peak of hydrogen atom in HNMR in Figure S13 for better understanding before acceptance.

After the authors address above issues, the manuscript can be accepted for publication.

Reviewer #3 (Remarks to the Author):

I am satisfied with the authors' response, and recommend this paper to be accepted.

Reviewer #2 (Remarks to the Author):

The paper is well explained. Most of the previous comments were addressed in the revised manuscript. Some minor comments should be considered:

1. The label fonts in the figures need to be unified.

Response: We are done.

2. There are still some grammatical errors in the article that need to be further carefully modified.

Response: Thanks for your suggestions. Following your suggestions, we have modified our manuscript.

3. Please show each peak of hydrogen atom in HNMR in Figure S13 for better understanding before acceptance.

Response: Thanks for your suggestions. Following your suggestions, we have done.

After the authors address above issues, the manuscript can be accepted for publication.